# RORγt inverse agonist TF-S14 inhibits Th17 cytokines and prolongs skin allograft survival in sensitized mice
Ahmed Fouda [1,2,3] ✉, Mohamed Taoubane Maallah[1,2,3], Araz Kouyoumdjian[2,3,4], Sarita Negi[2], Steven Paraskevas [1,2,3,4] & Jean Tchervenkov [1,2,3,4] ✉

Chronic antibody mediated rejection (AMR) is the major cause of solid organ graft rejection. Th17 contributes to AMR through the secretion of IL17A, IL21 and IL22. These cytokines promote neutrophilic infiltration, B cell proliferation and donor specific antibodies (DSAs) production. In the current study we investigated the role of Th17 in transplant sensitization. Additionally, we investigated the therapeutic potential of novel inverse agonists of the retinoic acid receptor-related orphan receptor gamma t (RORγt) in the treatment of skin allograft rejection in sensitized mice. Our results show that RORγt inverse agonists reduce cytokine production in human Th17 cells in vitro. In mice, we demonstrate that the RORγt inverse agonist TF-S14 reduces Th17 signature cytokines in vitro and in vivo and leads to blocking neutrophilic infiltration to skin allografts, inhibition of the B-cell differentiation, and the reduction of de novo IgG3 DSAs production. Finally, we show that TF-S14 prolongs the survival of a total mismatch grafts in sensitized mice. In conclusion, RORγt inverse agonists offer a therapeutic intervention through a novel mechanism to treat rejection in highly sensitized patients.

Kidney disease is the ninth leading cause of death in high income countries. The disease was long considered a terminal illness until both dialysis and kidney transplantation became widely practiced worldwide. Either peritoneal dialysis or hemodialysis poses a considerable burden on the patient and requires a specialized laborious care[1]. Transplantation, on the other hand, is considered the treatment of choice for end stage kidney disease, as it decreases the mortality, the morbidity and health-care costs compared to dialysis[2]. In Canada and USA, patients with kidney disease can get a kidney donation either from living or deceased donors within 3–5 years waiting time. However, 16% of kidney transplantation candidates are highly sensitized as defined by a high calculated panel reactive antibody (cPRA >80%). These patients are at a higher risk of developing AMR. Highly sensitized patients stay on the waiting lists for longer periods until a perfect donor match is allocated[3]. Although, both cellular and humoral immunity can be involved in allograft rejection. T cell mediated cellular rejection is currently fully preventable, however, humoral or commonly known as AMR is difficult to prevent, even when the prevention of the acute form is achieved for a short period it cannot be maintained for long term[4]. Chronic AMR can progress in an indolent fashion resulting in a slow accumulation of small injuries to the donated organ over time, which in many instances cannot be predicted with current diagnostic methods[5]. This type of rejection is now considered the major cause of renal graft rejection[6,7]. An increase in Th17 phenotype was observed in transplantation patients with chronic allograft dysfunction[8]. This observation suggests a close relationship between Th17 activation and AMR progression; however, this relationship is not well investigated. Several studies showed that Th17 and Tfh17 cells induce proliferation of B cells, antibody production and antibody class switching directly through the action of IL17A inside the germinal centers in chronic inflammatory and autoimmune disease characterized by the production of autoantibodies[9–11]. In solid organ transplantation, apart from the Tfh17 model inside the germinal centers, tissue resident Th17 are believed to trigger the development of a tertiary lymphoid tissue following allo-transplantation, which mediate B-cell differentiation, alloantibody production and antibody class switching outside the germinal center[12,13]. Those findings highlight the pathogenic role of Th17 cells, IL17A and IL21 cytokines in AMR. Therapeutic targeting of IL17A and IL21 has been evaluated in clinical trials for a variety of inflammatory and autoimmune indications, and anti-IL17A monoclonal antibodies were recently approved for psoriasis.

[1]Division of Surgical and Interventional Sciences, Department of Surgery, McGill University, Montréal, QC H3G 1A4, Canada. [2]Research Institute of the McGill University Health Centre, Montréal, QC H3H 2R9, Canada. [3]McGill University Health Centre, Montréal, QC H4A 3J1, Canada. [4]Division of General Surgery, Department of Surgery, McGill University, Montréal, QC H3G 1A4, Canada. ✉e-mail: ahmed.fouda@mail.mcgill.ca; jean.tchervenkov@muhc.mcgill.ca

On the other hand, targeting RORγt the master regulator of Th17 cells, was suggested as new therapy for solid organ rejection[14]. Previous research has shown that RORγt is a critical transcription factor for the secretion of IL17A[15,16]. RORγt inhibitors are particularly interesting as they can be used as oral medications of favorable adverse effects profile to substitute currently used oral immunosupressants[14]. Several studies have investigated the role of IL17A+ cells and Th17 in graft rejection in partial or full antigenic MHC class I and MHC class II mismatch mouse models of transplantation which mimics human incompatible transplantation[17–19]. Prior sensitization to the donor antigens can be done by performing two successive transplantations to the same recipient from different mouse species such as two skin transplantations, splenocytes injection followed by skin transplantation or a skin transplantation followed by a vascularized organ transplantation[20–22]. In such models the rejection to the vascularized organ mimics AMR[20,22]. In the current study we evaluated the anti-rejection immunomodulatory effect of RORγt inverse agonist TF-S14 on Th17 mediated cellular and AMR response in a sensitized mouse model of skin transplantation[21,23]. In our model of sensitized mice, we pre-sensitized C57BL/6 mice to BALB/C donor antigens to develop DSAs to mimic highly sensitized patients with cPRA = 100%. We show that RORγt inverse agonists decrease the production of IL17A, IL21 and IL22 in vitro and in vivo. We also show that TF-S14 increases total mismatch skin graft survival in pre-sensitized mice, attenuates Th17 mediated leukocytic infiltration to the graft and decreases de novo production of IgG3 DSAs. These effects are beneficial and can be further investigated to provide a new therapy to combat rejection in highly sensitized transplantation candidates.

## Results
### Effect of RORγt inverse agonists on Th17, Th1 and Treg polarization of human PBMCs

To compare the PBMCs to Th17 transformation between two transplantation candidate categories based on cPRA classification, namely, the highly sensitized whose cPRA >80% and the non-sensitized transplantation candidates whose cPRA <20%, we cultured the PBMCs from these two classes and from healthy volunteers (n = 3-5 per category) under Th17 polarization condition and measured the fractions of RORγt^hi/, IL17A+/, IL21+/, and IL22+/CD4+ on day 0, 7, 10 and 16. On days 0 and 7 the differences in the percentages of the IL17A+/, IL21+/ or IL22+/CD4+ cells were not significant between the three groups. On the other hand, the percentage of RORγt^hi/CD4+ cells was higher in PBMCs derived from highly sensitized compared to non-sensitized transplantion candidates on day 7 (Fig. 1a). On day 10 and 16 in addition to RORγt^hi, both the IL17A+/ and IL21+/CD4+ fractions of CD4+ cells were higher in highly sensitized compared to either non-sensitized transplantation candidates or healthy volunteers (Fig. 1b, c). To evaluate the effect of RORγt inverse agonists on human PBMCs to Th17 polarization, three 2,3 derivatives of 4,5,6,7-tetrahydro-benzothiophene RORγt inverse agonists, that we described in a recent report, were selected based on their potency in IL17A inhibition[23]. The three compounds TF-S1, TF-S2 and TF-S14 showed similar results in cell free TR-FRET binding assays (Fig. 1d–f). TF-S1 and TF-S2 were tested at three doses for their effect on Th17 signature cytokines in a a 16-day human Th17 polarization. The addition of TF-S1 and TF-S2 at 15, 150 or 750 nM in the last 48 h of culture incubation reduced the fractions of IL17A+/ and IL21+/CD4+ compared to vehicle treated cultures. In addition, TF-S2 at 15, 150 or 750 nM reduced IL22+/CD4+ fraction compared to vehicle, while TF-S1 reduced IL22+/CD4+ fraction at 750nM only (Fig. 1g–i and Supplementary Fig. 1a). Cyclosporine A reduced the percentage of IL21+/CD4+ and increased the percentage of IL22+/CD4+ cells. These results confirm the findings that have been reported before[24,25]. On the other hand, dexamethasone reduced the fraction of IL21+/CD4+ cells which confirms recently reported findings about the effects of dexamethasone on Tfh cells and IL21 mRNA and protein expression[26]. Additionally, dexamethasone reduced IL17A+/ and IL22+/CD4+ fraction at only the highest concentration tested 750 nM (Fig. 1g, i). Further we tested the two compounds in Th1 and Tregs polarization assays to determine their effects on both cell

types. In a 3-days Th1 polarization experiment TF-S1 and TF-S2 had no effect on IFNγ+/CD4+ compared to dexamethasone or cyclosporine A (Fig. 1j and Supplementary Fig. 1b). In a 6-days Treg polarization assay the two compounds had no effect on percentage of Tregs when compared to the Treg inhibitor tacrolimus or the inducer sirolimus. The identification was based on fraction of CD127^lo CD25^hi FOXP3+/CD4+ cells (Fig. 1k and Supplementary Fig. 1c). Despite their superior effects in vitro, however the two compounds were not selected for further evaluation because of the poor stability of TF-S1 and poor solubility of TF-S2. Additional 2,3 derivatives of 4,5,6,7-tetrahydro-benzothiophene were further tested in Th17 polarization assay and cell free binding assays[23].

### Effect of RORγt inverse agonists on Th17 polarization of human PBMCS from highly sensitized kidney transplantation candidates

To evaluate the effect of TF-S2 and TF-S14 on RORγt expression and Th17 signature cytokines IL17A as well as IL21 and IL22 expression in Th17 polarized PBMCs from highly sensitized kidney transplantation candidates, we added either TF-S2, TF-S14 or GSK2981278 reference RORγt inverse agonist to 14-days Th17-polarized human PBMCs from 7 transplantation candidates who had a cPRA >95%. The RORγt inverse agonists were added to the culture media at a concentration of 15 nM for 48 h. On day 16, following a 5-h PMA/ionomycin stimulation, we measured the fractions of RORγt^hi/, IL17A+/, IL21+/ and IL22+/CD4+ cells. The RORγt inverse agonists GSK2981278, TF-S2 and TF-S14 reduced the RORγt protein expression in CD4+ T cells compared to the vehicle control as measured by mean fluorescence intensity (MFI) (Fig. 2a and Supplementary Fig. 2a). In addition, all the tested RORγt inverse agonists reduced the percentages of RORγt^hi/, IL17A+/, IL21+/ and IL22+/CD4+ in Th17-polarized human PBMCs compared to vehicle control (Fig. 2b–f and Supplementary Fig. 2b). Moreover, to determine the effect of the RORγt inverse agonists on non-Th17 signature cytokines, we measured the percentage of IFNγ+/CD4+ and no difference was found between vehicle control and RORγt inverse agonists treated Th17-polarized human PBMCs (Fig. 2f). However, the ratio of Th17/Th1 (IL17A+CD4+/ IFNγ+CD4+ cells) was reduced in RORγt inverse agonists treated compared to vehicle treated Th17-polarized human PBMCs control cultures (Fig. 2g). Finally, RORγt inverse agonists decreased the concentration of the secreted IL17A and IL21 cytokines in the culture media of Th17-polarizaed human PBMCs collected on day 16 following PMA/ ionomycin stimulation (Fig. 2h, i).

### RORγt inverse agonist TF-S14 prolongs the survival of skin allograft in BALB/C-sensitized C57BL/6 mouse model

To evaluate the effect of RORγt inverse agonist TF-S14 in vivo, we developed a BALB/C-sensitized C57BL/6 mouse, hereafter referred to as "sensitized", model of complete mismatch full thickness BALB/C skin transplantation. In this model three IP doses of BALB/C cell suspension were given over a 2-week period to C57BL/6 mice before the transplantation with a BALB/C mouse dorsal skin graft (Fig. 3a). To validate the model, we performed a Th17 polarization of T cells derived from the sensitized mice. The percentages of RORγt^hi/, IL17A+/, IL21+/ and IL22+/CD4+ were higher in BALB/C-sensitized compared to non-sensitized C57BL/6 mice (Fig. 3b–e and Supplementary Fig. 3). In addition, we evaluated the effect of adding TF-S14 (15 nM) on Th17 polarization of T cells isolated from sensitized and non-sensitized mice. TF-S14 decreased the percentages of RORγt^hi/, IL17A+/, IL21+/ and IL22+/ CD4+ in Th17 polarized T cells isolated from the splenocytes of sensitized mice compared to vehicle control (Fig. 3b–e). Similarly, TF-S14 decreased the percentages of IL17A+/ and IL21+/CD4+ in Th17 polarized T cells from non-sensitized mice compared to vehicle control (Fig. 3c, d and Supplementary Fig. 3). In vivo TF-S14 IP administration to sensitized mice transplanted with BALB/C skin grafts, at a daily dose of 1 mg/kg alone or in combination with tacrolimus 0.5 mg/kg improved the surgical wound inflammation ASEPESIS score for mice[27] compared to no treatment (NT) or tacrolimus 0.5 mg/kg treatment alone, respectively (Fig. 3f,

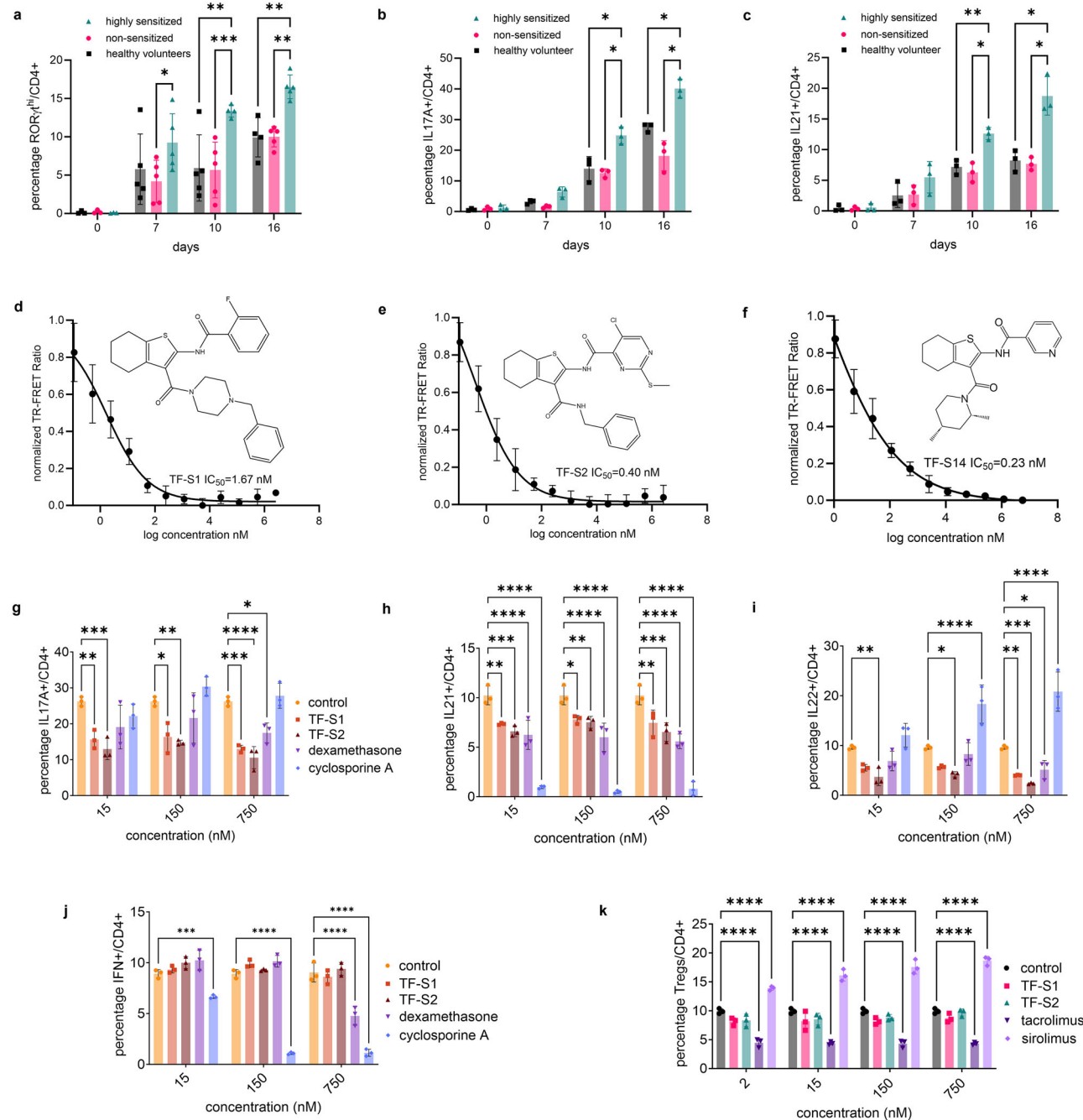

**Fig. 1 | In vitro Th17 polarization time course of human PBMCs from highly sensitized kidney transplantation candidates and the effect of RORγt inverse agonists on cytokine production from Th17, Th1 and Treg cells.** PBMCs from highly sensitized, non-sensitized transplantation candidates and healthy volunteers were polarized for 0, 7, 10, 16 days then stimulated with PMA/ionomycin/monensin/prefoldin cocktail for 5 h. The percentages of RORγt$^{hi}$/ (**a**), IL17A+/ (**b**) and IL21+/CD4+ cells (**c**) were measured by flow cytometry. Dose-response curves of TR-FRET assay of RORγt-LBD: serial dilutions of compounds TF-S2, TF-S10, TF-S14 were incubated with 1.5 nM GST−RORγt (LBD); 90 nM biotinylated RIP140 coactivator peptide; 50 nM streptavidin-APC; 1.5 nM Eu-anti GST. The data are represented as percentage TR-FRET ratio. IC$_{50}$ of TF-S1, TF-S2 and TF-S14 were 1.67, 0.40 and 0.23 nM, respectively (**d**–**f**). Effect of TF-S1, TF-S2, cyclosporine A and dexamethasone on Th17 polarization: the percentages of IL17A+/ (**g**), IL21+/ (**h**) and IL22+/CD4+ cells (**i**) were measured by flow cytometry. Effect of TF-S1, TF-S2, cyclosporine A and dexamethasone on human PBMCs to Th1 polarization: the percentage of IFNγ+/CD4+ cells was measured on day 3 by flow cytometry (**j**). Effect of TF-S1, TF-S2, sirolius and tacrolimus on human PBMCSs to Treg polarization: the percentage of CD127$^{lo}$CD25$^{hi}$FOXP3+/CD4+ was measured on day 6 by flow cytometry (**k**). Data are presented as mean ± SD, *p < 0.05, **p < 0.01, ***p < 0.005, ****p < 0.001, two-way ANOVA and Tukey's test, n = 3–5 per group.

g). In another experiment, the histological evaluation of BALB/C skin grafts that were isolated on day 5 post-transplantation and stained with hematoxylin and eosin, revealed that daily treatment of the sensitized mice with IP injections of RORγt inverse agonist TF-S14 at doses of 1, 10 or 100 mg/kg alone or in combination with tacrolimus 0.5 mg/kg resulted in the reduction or absence of neutrophilic infiltration to the skin grafts

compared to a diffuse neutrophilic infiltration in BALB/C skin grafts isolated from NT sensitized mice or those treated daily with tacrolimus at a dose of 0.5 mg/kg following the transplantation surgery (Fig. 3h). Furthermore, the daily IP injection of TF-S14 at doses of 1, 10 or 100 mg/kg prolonged the graft median survival time (MST) in sensitized mice to 13.5, 15 and 15 days, respectively, compared to a graft MST of 6 days in NT

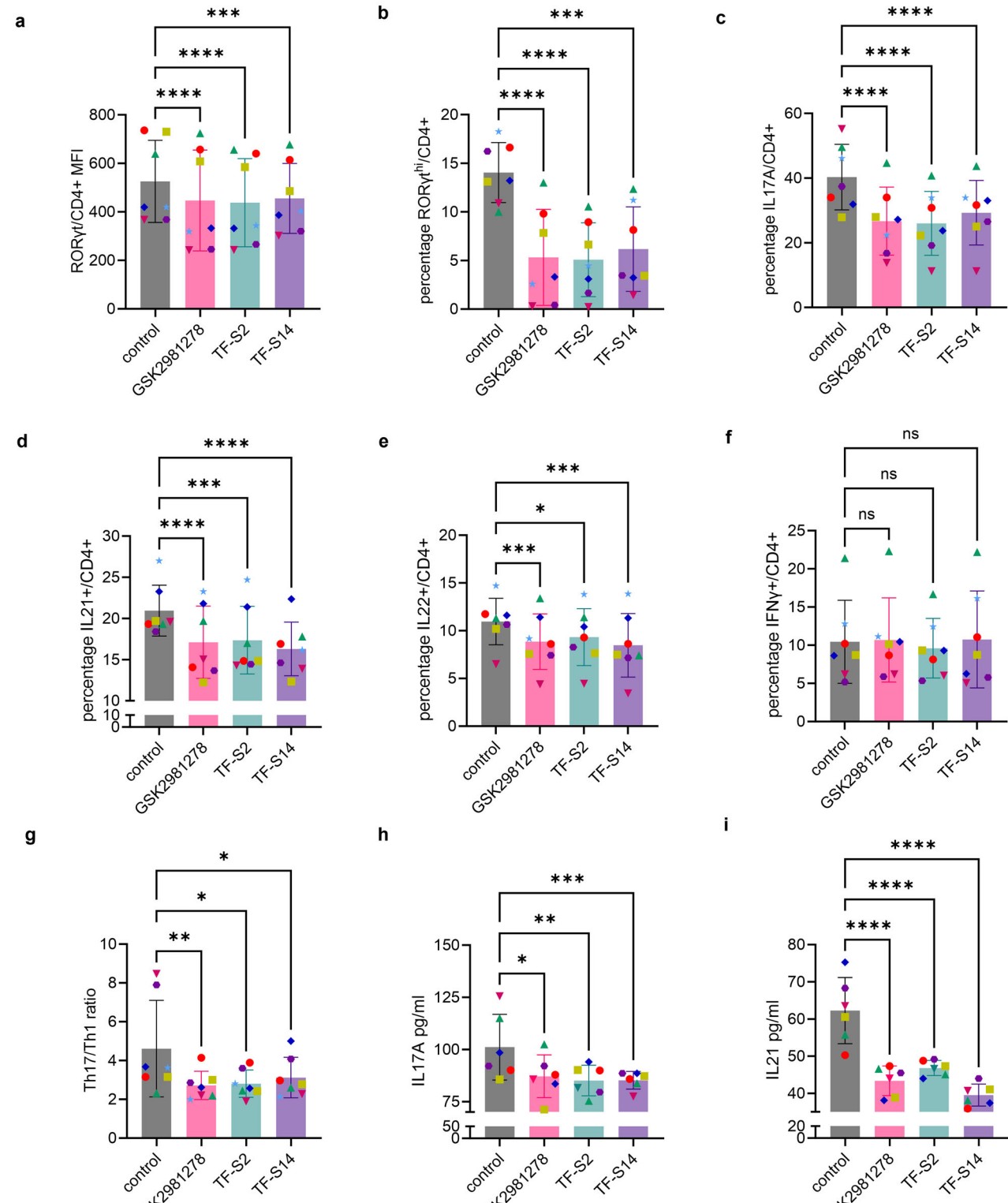

**Fig. 2 | Effect of RORγt inverse agonists TF-S2 and TF-S14 on Th17 polarization of human PBMCs collected from highly sensitized transplantation candidates.** PBMCs from highly sensitized kidney transplantation candidates were treated with CD3/CD28 magnetic beads for 16 days in presence of IL6 10 ng/ml, IL1β 10 ng/ml, TGFβ 10 ng/ml, and IL23 10 ng/ml. Culture medium was changed every 3 days for the whole duration. GSK298128 (15 nM), TF-S2 (15 nM), TF-S14 (15 nM) or vehicle were added to the polarized cells on day 14 for 48 h and at the end of incubation cells were stimulated with PMA/ionomycin/monensin/prefoldin cocktail for 5 h.

RORγt$_{BV650}$ MFI of CD4+ cells (**a**), the percentages of RORγt$^{hi}$/ (**b**), IL17A+/ (**c**), IL21+/ (**d**), IL22+/ (**e**) and IFNγ+/CD4+ (**f**) were measured with flow cytometry. The ratio of Th17/Th1 was measured by dividing the percentage of IL17+/CD4+ by the percentage of IFNγ+/CD4+ cells (**g**). The concentration of IL17A and IL21 secreted in the culture media were both measured in supernatant using ELISA (**h**, **i**). Data are presented as mean ± SD, *$p < 0.05$, **$p < 0.01$, ***$p < 0.005$, ****$p < 0.001$, two-way ANOVA and Tukey's test, $n = 6$–7 per group.

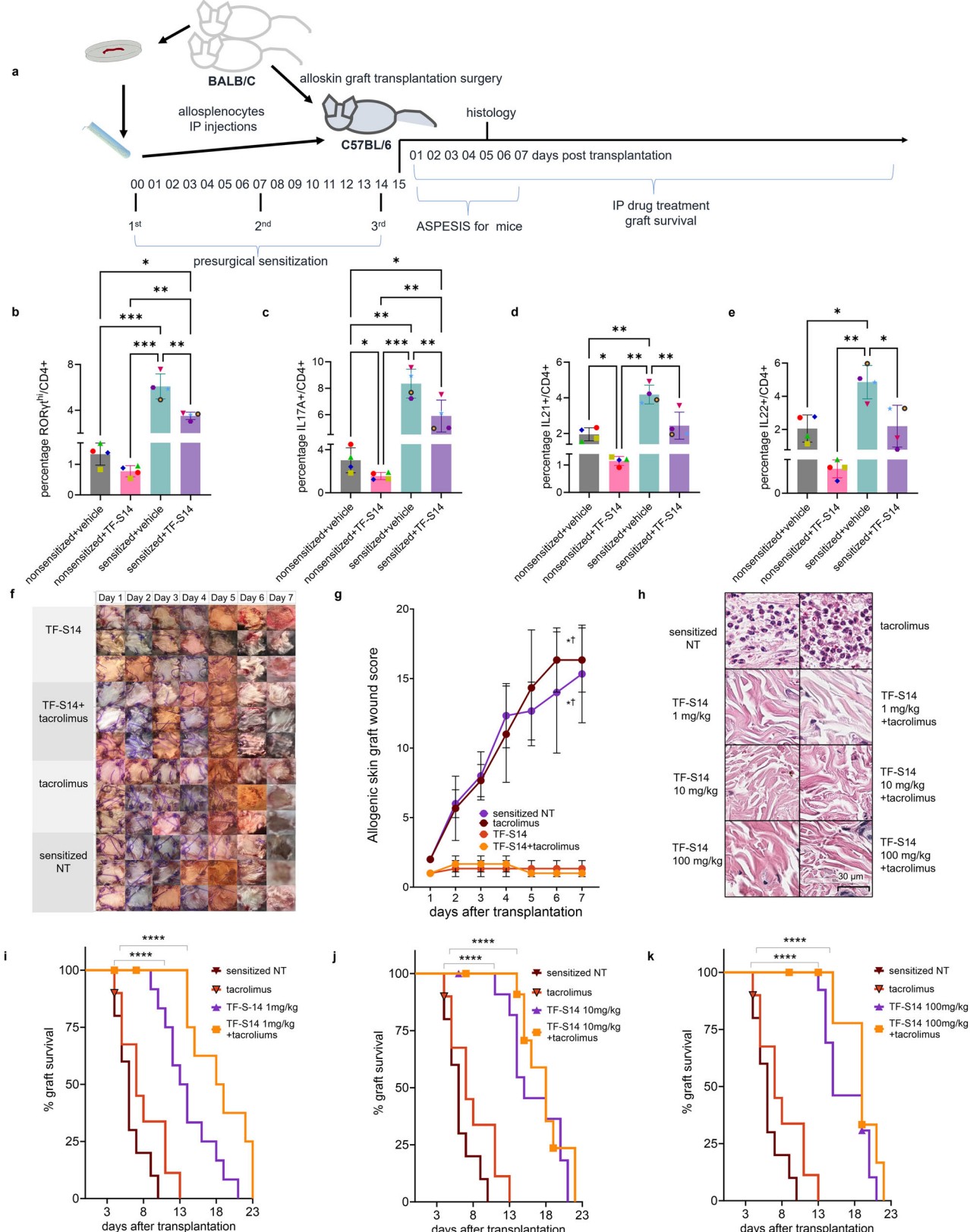

sensitized mice group. Similarly, the daily administration of the combination of TF-S14 at doses of 1, 10 or 100 mg/kg with tacrolimus 0.5 mg/kg increased the graft MST in sensitized mice to 18.5, 18 and 19 days, respectively, compared to a graft MST of 7 days in tacrolimus 0.5 mg/kg only treated sensitized mice group (Fig. 3i–k).

## RORγt inverse agonist TF-S14 decreases IL17A protein expression in spleen and tissue resident leukocytes

To measure the effect of TF-S14 on the immune system following daily administration to transplanted sensitized mice. Th17/Tfh17 signature cytokine protein expression was measured in splenocytes from transplanted

**Fig. 3 | Effect of RORγt inverse agonist TF-S14 on sensitized mice T-cell to Th17 polarization, allograft survival and histological features of full thickness BALB/C skin transplantation to BALB/C-sensitized C57BL/6 mice.** C57BL/6 were sensitized by three IP injections of BALB/C splenocytes ($10^7$) on day 0, 7 and 14. On day 15 mice were transplanted with $12 \times 12$ mm BALB/C dorsal skin grafts (**a**). The sensitization was assessed by polarizing T cells isolated from splenocytes of non-sensitized and sensitized mice to Th17 phenotype for 4 days in the presence or absence of TF-S14 (15 nM), followed by stimulation with PMA/ionomycin/monensin/prefoldin cocktail for 5 h. The percentages of RORγt^hi/CD4+ (**b**), IL17A+/CD4+ (**c**), IL21+/CD4+ (**d**) and IL22+ (**e**) were measured by flow cytometry. Data are presented as mean ± SD, *$p < 0.05$, **$p < 0.01$, ***$p < 0.005$, REML and Tukey's tests, $n = 4$ per group. Following skin BALB/C allograft surgery, pre-sensitized C57BL/6 mice were assigned into sensitized no treatment (NT), tacrolimus (0.5 mg/kg), TF-S14 (1mg/kg) or TF-S14 (1mg/kg) and tacrolimus (0.5 mg/kg) combination daily treatment groups. The allografts were inspected daily, and area ($12 \times 12$ mm) were imaged with 12 megapixels camera (**f**). The graft area was scored for swelling (1–5), redness (1–5), presence of serous exudate (1–5), presence of purulent exudate (2–10) and loss of hair (2–10) to determine ASPESIS wound score for mice over time. Data are presented as mean ± SD, *†$p < 0.05$ compared to sensitized NT and tacrolimus treated groups, respectively, Friedman one-way ANOVA and Dunn's test, $n = 3$ per group (**g**). Histopathology photomicrographs of skin allografts isolated on day 5 post-transplantation (H&E, ×40) show that grafts from mice treated with TF-S14 1, 10 or 100 mg/kg alone or in combination with tacrolimus 0.5 mg/kg have low neutrophilic infiltration in <10% of graft area compared to high neutrophilic infiltration (>10%) in sensitized NT mice (**h**), $p < 0.05$, Kruskal–Wallis one-way ANOVA and Dunn's test, $n = 3$–4 per group. Graft survival assessed daily until complete necrosis or graft loss (**i–k**). Data are presented in Kaplan–Meier survival curve, ****$p < 0.001$, Log-rank (Mantel–Cox) test, $n = 8$–12 per group.

sensitized mice treated daily with IP injection of TF-S14 at doses of 1, 10 or 100 mg/kg alone or in combination with tacrolimus 0.5 mg/kg using flow cytometry following a 5-h PMA/ionomycin activation. The percentage of IL17A+/CD4+ cells were reduced in all TF-S14 treated compared to NT controls. There was no difference between the three doses tested (1, 10 and 100 mg/kg) on the IL17A+/CD4 fraction in spleen. Similarly, the combination of TF-S14 at doses of 1, 10 or 100 mg/kg with tacrolimus 0.5 mg/kg, administered daily following transplantation surgery, decreased IL17A+/CD4+ fraction in splenocytes from combination treated compared to tacrolimus 0.5 mg/kg treated sensitized mice (Fig. 4a, b). In addition, RORγt inverse agonist TF-S14 treatment reduced the expression of IL17A and RORγt in Ly6G+ neutrophils and Ly6G+ neutrophil like monocytes (Fig. 4c–f). At the graft level, there was a decrease in the infiltration of Ly6G+ neutrophils in the grafts that were collected on day 5 from the mice treated with RORγt inverse agonist TF-S14 at doses of 10 and 100 mg/kg alone or in combination with tacrolimus 0.5 mg/kg compared to NT controls (Fig. 4g, h). Similarly, TF-S14 treatment at doses of 10 and 100 mg/kg alone or in combination with tacrolimus 0.5 mg/kg reduced IL17A+ lymphocytic infiltration to the grafts compared to those isolated from NT sensitized mice (Fig. 4i, j). Neither the neutrophilic nor lymphocytic infiltration in tacrolimus only treated mice was different from those infiltrations seen in NT controls (Fig. 4g–j). Th17 and IL17A+ lymphocytes were smaller in size and cytoplasmic IL17A granules were less dense compared to those found in the grafts isolated from NT or tacrolimus 0.5 mg/kg treated sensitized mice (Supplementary Fig. 4a, b). In TF-S14 plus tacrolimus treated mice, the skin grafts that survived beyond 13 days showed thinning of hypodermis and fibrosis of dermis and epidermis (Supplementary Fig. 4c). IL17A expression, in CD169+ macrophages and CD169+ macrophage like monocytes, was reduced in the splenocytes isolated from TF-S14 treated compared to NT or tacrolimus treated sensitized mice (Fig. 4k, l).

## RORγt inverse agonist TF-S14 decreases antibody class switching, B cell differentiation and de novo IgG3 DSAs

To determine the effects of RORγt inhibition on B cells. We measured both IgM and IgG3 immunoglobulins to determine the effect of RORγt inverse agonists administration of antibody class switching from IgM to IgG3. This was based on previous reports that showed that Th17 promotes IgG1, IgG2a, IgG2b and IgG3 subtypes production in B cells, while IL17A specifically promotes IgG2 and IgG3 subtypes[9]. Our results show that TF-S14 1, 10, 100 mg/kg treated mice had lower IgG3+IGM+/CD19+CD138+ mature B cells compared to NT sensitized or tacrolimus 0.5 mg/kg treated sensitized mice (Fig. 5a, b). In addition, TF-S14 treatment at doses of 10 mg/kg and 100 mg/kg reduced the percentage of IgM^hi/CD19+CD138+ cells, compared to NT sensitized mice (Fig. 5a, c). Moreover, TF-S14 treatment to sensitized mice decreased serum IgG3 content (Fig. 5d). However, the treatment didn't decrease the total IgG concentration in mice sera in TF-S14 treated compared to NT sensitized mice, however sensitized mice from all groups had high concentrations of total IgG compared to non-sensitized mice (Fig. 5e). The IgM to IgG3 class switching was observed in the spleen sections using immunofluorescence staining of IgM+, IgG+ and ly6G+ cells (Fig. 5f). There was a decrease in IgG3+ cell clusters which was confirmed by the reduction of IgG3_Alexa Fluor 594 MFI in the spleen section from TF-S14 treated sensitized mice groups compared to vehicle or tacrolimus treated groups (Fig. 5g).

## RORγt inverse agonist TF-S14 decreases ILC3 fraction in splenocytes of sensitized mice

Previous work showed that $Rorc^{-/-}$ mice had fewer splenic Lin−CD117+CD127+ILC3 compared to $Rorc^{+/+}$ (wild-type) mice[28]. To determine the effect of RORγt inhibition on ILC3 we compared the percentage of CD117+CD127+/Lin− population in TF-S14 treated, tacrolimus treated, NT sensitized and non-sensitized mice. The ILC3/Lin− fraction was higher in tacrolimus treated compared to NT sensitized mice. RORγt inverse agonist TF-S14 alone or in combination with tacrolimus reduced the ILC3 fraction compared to tacrolimus 0.5 mg/kg only treated sensitized mice and restored the ILC3 levels to the normal levels seen in non-sensitized mice (Fig. 5h, i).

## Discussion

The role of Th17 and other IL17A producing cells in heart and skin graft rejection have been deliberated before in many studies[19,29–31]. However, the correlation between prior sensitization state and leukocyte predisposition to transformation to IL17A+ phenotypes was not studied before. We showed in a preliminary report that Th17 phenotype is present in the peripheral blood of highly sensitized kidney transplantation candidates[32]. In the current study, we found a positive correlation between prior sensitization status and IL17A+CD4+ T cells in an in vitro Th17 polarization of human PBMCs. Previous studies have shown that Th17 is involved in both cellular rejection and AMR[29,30]. The hallmark of Th17 mediated cellular rejection is the acute massive neutrophilic infiltration of the grafts[18], which we observed during graft rejection in sensitized mice. This is mediated through IL17A and other cytokines secreted from Th17[13,18]. Our findings show that in addition to its production in T cells, IL17A was also produced by other leukocytes such as neutrophils and macrophages as we found in the splenocytes of sensitized mice. Both neutrophils and macrophages have been shown to produce IL17A[33,34]. The increase in IL17A+ neutrophils and macrophages in sensitized mice compared to non-sensitized mice indicates intercellular positive feedback where the IL17A and other cytokines that are secreted from Th17 cells promote an increase in RORγt expression in different leukocytes. Subsequently this positive feedback leads ultimately to an increase in IL17A expression in these leukocytes. This proposed mechanism can be explained by the reduction of RORγt+ neutrophils that we have observed following the administration of RORγt inverse agonists daily treatment with three doses of TF-S14. The positive feedback of IL17A and other Th17/Tfh17 cytokines has been reported to work in both autocrine and paracrine fashion[35]. The second mechanism of graft rejection involving Th17 help, is the promotion of tertiary lymphoid tissue formation that requires IL21 cytokine secretion, which was reduced by RORγt inverse agonists[12,13]. Our findings show that the inverse agonists of RORγt decrease

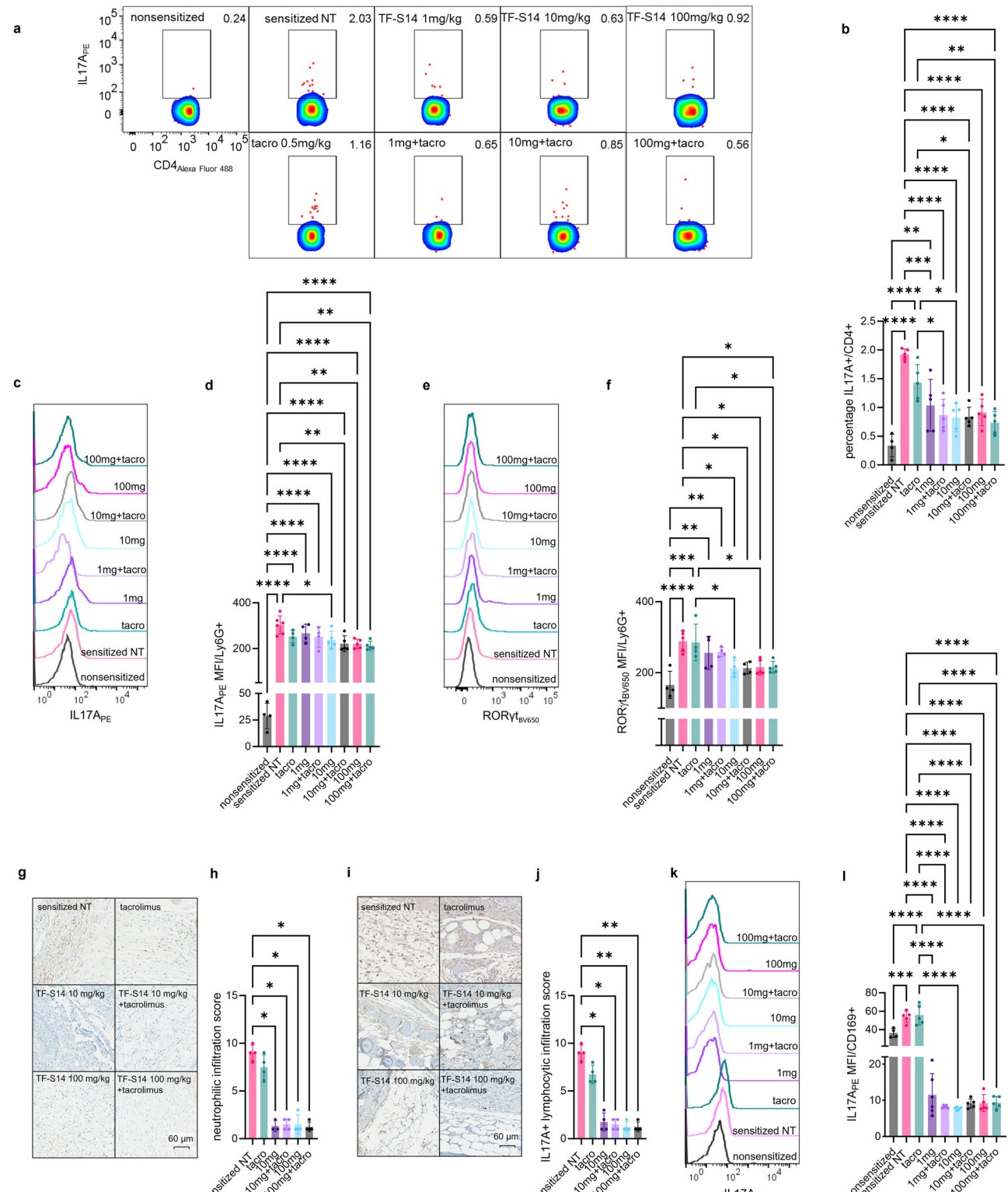

**Fig. 4 | Effect of RORγt inverse agonist TF-S14 on IL17A production in splenic CD4+ T cells, CD169+ macrophages, Ly6G+ neutrophils and graft infiltrating leukocytes following full thickness BALB/C skin transplantation to BALB/C-sensitized C57BL/6 mice.** Splenocytes were isolated from NT sensitized control, tacrolimus 0.5 mg/kg (tacro), TF-S14 1, 10 or 100 mg alone and or in combination with tacrolimus 0.5 mg/kg treated mice on the day of rejection. The splenocytes were activated with PMA/Ionomycin/monensin/prefoldin cocktail and percentage of IL17A+/CD4+ (**a, b**), RORγt$_{BV650}$ MFI (**c, d**), IL17A$_{PE}$ MFI of Ly6G+ cells (**e, f**) were measured by flow cytometry. Data are presented as mean ± SD, *$p < 0.05$, **$p < 0.01$, ***$p < 0.005$, ****$p < 0.001$, one-way ANOVA and Tukey's test, $n = 4$–6

per group. Histopathology photomicrographs show BALB/C skin allografts ($n = 3$–4 per group) that were isolated on day 5 post-transplantation and stained with anti-ly6G (brown), ×40 (**g, h**) or anti-IL17A (brown), ×40 (**i, j**). The percentage area of Ly6G+ neutrophilic and IL17A+ lymphocytic infiltrations were scored from 1 (<10%) to 10 (>90%) of the graft area. Data are presented as mean ± SD, *$p < 0.05$, **$p < 0.01$, Kruskal–Wallis one-way ANOVA and Dunn's test, $n = 3$–4 per group. IL17A$_{PE}$ MFI in CD169+ cells was measured by flow cytometry (**k, l**). Data are presented as mean ± SD, *$p < 0.05$, **$p < 0.01$, ***$p < 0.005$, ****$p < 0.001$, one-way ANOVA and Tukey's test, $n = 4$–6 per group.

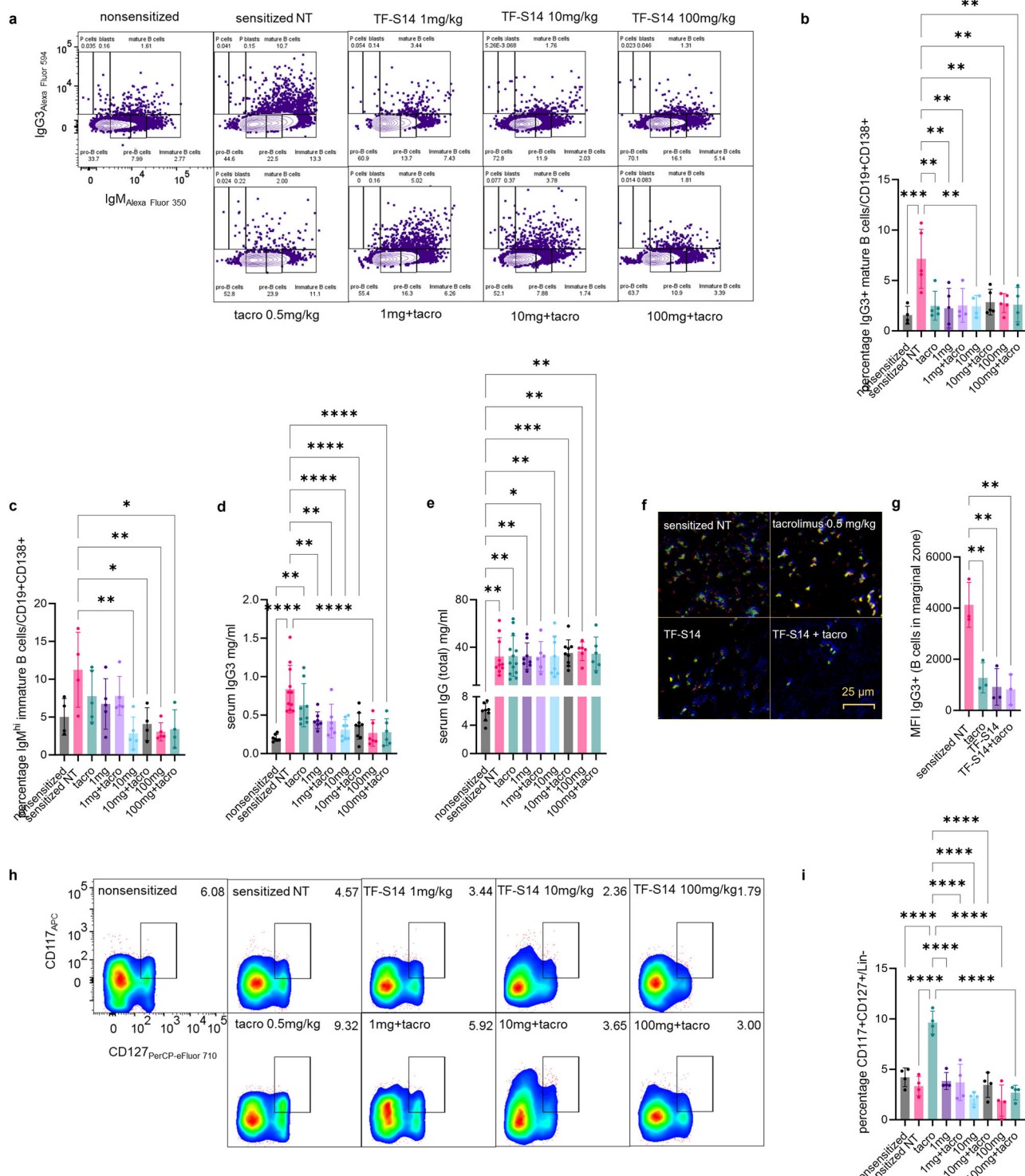

**Fig. 5 | Effect of RORγt inverse agonist TF-S14 on B-cell differentiation, IgM to IgG3 class switching, ILC3 fraction, serum IgG/IgG3 content following full thickness BALB/C skin transplantation to BALB/C-sensitized C57BL/6 mice.** Splenocytes were isolated from sensitized NT control, tacrolimus 0.5 mg/kg (tacro), TF-S14 1, 10 or 100 mg alone and combination treated mice on the day of rejection and activated using anti-IgM/anti-IgG3. The percentage of mature B cells IgG3+IgM+/CD19+CD138+ and immature B cells IgM$^{hi}$IgG3−/CD19+CD138+ were measured by flow cytometry (**a**–**c**). The content of IgG and IgG3 were measured in mice sera with ELISA (**d**, **e**). Photomicrographs show spleen sections that were stained with anti-IgG3 (red), anti-IgM (blue), anti-Ly6G (green), showing increase in IgG3 cells in the spleens in vehicle or tacrolimus treated mice compared to TF-S14 alone or in combination with tacrolimus 0.5 mg/kg (**f**, **g**). Splenocytes were stained with ILC3 phenotyping surface markers and the percentage of CD117+CD127+/ Lin− cells was measured by flow cytometry (**h**, **i**). Data are presented as mean ± SD, *$p < 0.05$, **$p < 0.01$, ***$p < 0.005$, ****$p < 0.001$, one-way ANOVA and Tukey's test, $n = 3$–13 per group.

IL21 expression in ROR+ T cells in vitro human PBMCs polarization, however we did not detect the difference in the splenocytes isolated from transplanted mice. An explanation for this is that IL21 expression can be correlated to RORγt[hi] CD4+ rather than to RORγt+CD4+ cells, which indicates that IL21 expression in Th17 is a subsequent stage to IL17A production in peripheral RORγt+ cells and as well influenced by IL17A positive feedback loop. Previous reports have shown that IL21 works as an autocrine factor that promotes or sustains the Th17 phenotype[36]. In our human in vitro Th17 polarization experiment, we found that IL21 secretion correlates with RORγt[hi] expression in Th17 which indicates a sustained Th17 phenotype.

The third mechanism by which Th17 cells mediate graft rejection is the through the promotion of antibody class switching which in addition to IL17A also requires IL21 and preferentially promotes IgG2a and IgG3 classes[9,13]. Although we did not see a difference in IL21+/CD4+ fraction in splenocytes from the mice that were treated with TF-S14; however, we observed a reduction in IgG3 serum level and a reduction in IgM to IgG3 antibody class switching which can be both correlated to the reduction of IL21 expression in RORγt+CD4+ cells following RORγt inverse agonist treatment in mice. We can confirm this effect on IL21 production based on our observations in human PBMCs Th17 polarization and observations from other studies[37]. Previous research identified that de novo alloantibodies produced during rejection are of IgG1 and IgG3 subtypes. These subtypes have strong complement binding capacity and are associated with acute AMR[38]. Both IgG1 and IgG3 subclasses were shown to be associated with acute and chronic AMR worse outcomes[39,40]. In addition to those three mechanisms, as we previously mentioned, we found that sensitization in mouse model promotes RORγt+ neutrophils and macrophages. Both cell types are considered among the professional antigen presenting cells. They are mainly known to be present antigens to T cells, but they also present antigens to B cells[41,42]. We observed an increase in neutrophil and macrophage B cell help in the spleen sections of sensitized mice and we found that they were both reduced in the mice that were treated daily with TF-S14. This can be a fourth mechanism by which RORγt inverse agonists decrease antibody class switching in B cells through the reduction of leukocytes recruitment at the inflammation site. This effect on leukocytes infiltration was observed in the skin allografts as a reduction or complete absence of neutrophils in the grafts of the mice treated with TF-S14. This effect can also explain the ability of the higher doses of TF-S14 to block the differentiation of pre-B cells to immature IgM[hi] B cells.

Moreover, the daily treatment with the RORγt inverse agonist TF-S14 resulted in restoring ILC3 fraction in sensitized mice to near the normal levels found in non-sensitized mice which were, on the other hand, higher than normal in tacrolimus only treated mice. Previous research has shown that the percentage of ILC3 is reduced in patients undergoing acute cellular organ rejection compared to no rejection kidney transplantation patients[43]. Even though the reduction of ILC3 that we observed in sensitized mice compared to non-sensitized mice was not significant, this is still aligned with previously reported findings[43]. Other studies have shown that the increase in the ILC3 indicates a Th2 minimal persistent inflammation[44]. This can explain the increase in the percentage of ILC3 in tacrolimus only treatment mice as it has previously demonstrated that tacrolimus decreases the Th1/Th2 ratio[45]. ILC3 are non-specific innate immune cells which like other non-specific innate immune cells can be short lived first responders which explains both their exhaustion in active rejection and the increase in their percentage when the immune system is triggered in a minimal chronic fashion. The ability of the RORγt inverse agonist TF-S14 to restore the normal levels of ILC3 cells indicates an excellent control of the inflammation at the graft site. This effect can be mediated through the reduction of IL22 cytokine expression in RORγt+ cells. Like IL21, we found that IL22 production in Th17 cells is correlated with the percentage of RORγt[hi] rather than RORγt+ and can eventually be indirectly influenced by IL17A positive feedback loop trough RORγt. It has been shown that ILC3 express RORγt and they produce both IL17A and IL22 through which they mediate their innate immune function and eventually perform some functions of Th17

cells such as mucus protection against pathogens[46]. We conclude that RORγt inverse agonist treatment prevents the exhaustion of ILC3 and reduces their recruitment at the inflammatory site as well and eventually does not result in an elevation of ILC3 overall percentage which can be a useful marker of successful RORγt inhibition. On the other hand, studies showed that ILC3 cells play a role in T-cell independent production of IgM, IgG2a and IgG3. In the current study both the restoration of normal levels of ILC3 and reduction of serum IgG3 levels was observed in the mice treated with TF-S14 compared to tacrolimus only treated mice.

Finally, we found that TF-S14 inverse agonist treatment prolonged, while tacrolimus treatment did not influence graft survival in sensitized mice as it is not as effective in blocking Th17 response which is a major player in hyperacute cellular, and AMR encountered in this model[30,45,47]. This is because the Th17 response that develops in the accelerated rejection mouse model starts as early as 3 days post transplantation, which has been shown to occur in mouse model of renal and skin transplantation[17,31]. In fact, following a three BALB/C splenocyte IP injections sensitization protocol, memory Th17, memory B cells and preformed DSAs develop in the sensitized mice before the skin transplantation surgery as demonstrated in previous studies[17,22,31]. The subsequent events such as the neutrophilic infiltration that is the predominant end stage mechanism in the accelerated skin allograft rejection model can typically occur early at 24 h post transplantation and reach the peak within the first 6 days[21]. On the other hand, the Th1 or IFNγ mediated cellular response typically peaks at 10–12 days post transplantation as seen in non-sensitized mouse model of skin transplantation and full rejection occurs at 18 days[48]. This onset of IFNγ response seems to be unchanged in sensitized mouse model of transplantation as the median survival that we observed in RORγt inverse agonist treated mice was at 18 days which matches with Th1 rejection timeline rather than 6 days that we concluded to be the Th17 hyperacute rejection timeline which is characterized with massive neutrophilic infiltration. Even though tacrolimus is effective in blocking Th1 responses, however its effect on Th17 is marginal and in fact it may potentiate Th17 response on long term use[30,45,47]. Dysregulation of Th17 response can be mediated through its effect on Th1/Th2 or the reduction of Treg population[30,45]. The histological image of the skin grafts from the mice treated with TF-S14 in combination with tacrolimus that survived beyond 13 days showed resemblance to Th2 mediated scleroderma or eosinophilic fasciitis. This indicates that with successful inhibition of Th1 and Th17 responses by tacrolimus and RORγt inverse agonist TF-S14, respectively, a third mechanism mediated through Th2 orchestrates the rejection. Previous research has shown that Th2 mediated chronic rejection develops in mice deficient in RORγt and T-bet[49]. Further research is required to investigate Th2 mediated rejection mechanisms that develop under dual pharmacologic inhibition of both Th1 and Th17. In addition, further studies are required to evaluate the effects of TF-S14 on memory Th1 and CD8+ T cells and other immune cells activated during acute cellular rejection in a non-sensitized allogenic transplantation model. In such studies, the use of a non-sensitized mouse model is crucial to avoid the neutrophilic infiltration resulting from the early activation of memory Th17 response. This infiltration masks memory Th1 and CD8+ T-cell response.

In conclusion, we found that the RORγt inverse agonist TF-S14 is effective in improving graft survival by blocking four processes namely leukocyte recruitment, tertiary lymphoid tissue formation, B cell differentiation and de novo IgG3 DSAs production. These effects are mediated through the inhibition of RORγt in Th17 and other leukocytes such as neutrophils, macrophages and ILC3. These effects are immediate and long term, accordingly, TF-S14 can be used to prevent or treat solid graft rejection.

## Methods
### Human blood samples collection
Blood samples collection protocol (12-075-GEN) was approved by the Research Ethics Board of the Research Institute of the McGill University Health Centre. Written consents were received from the participants before inclusion in the study. Blood samples were collected from healthy donors

and kidney transplantation candidates on dialysis by a registered nurse. Samples were all coded and identified by number.

## Human Th17 polarization

Peripheral blood mononuclear cells were isolated by density gradient centrifugation. Whole blood was diluted with 1x PBS and then layered gently over 14 ml of lymphocyte separation medium in a 50 ml centrifuge tube and centrifuged for 30–40 min at $400 \times g$. PBMCs layer and is a characteristic white and cloudy "blanket" was gently removed using a 10 ml pipette and added to warm medium or PBS (1:3) to wash off any remaining platelets and centrifuged at $400 \times g$ for 10 min. The pelleted cells were counted, and the percentage viability estimated using trypan blue staining. PBMCs were resuspended to $5 \times 10^6$ cells/ml in freezing medium containing 10% DMSO and 90% fetal bovine serum (FBS) and were placed inside a freezing container at $-80\,°C$ overnight to allow gradual and even cooling. The following day, samples were moved to a liquid nitrogen tank for long-term storage. In total, 500 ml bottle of RPMI at 4 °C was opened in a biological safety cabinet. Half of the media was transferred to a 500 ml 0.22 um bottle top filter (with the vacuum turned off). In total, 59 ml filtered heat inactivated FBS, 6 ml penicillin-streptomycin solution (100x), 6.0 ml of L-glutamine (100x), 6.0 ml MEM non-essential amino acids 100x, 6.0 ml of sodium pyruvate were added, and the volume was brought up to 500 ml by adding the RPMI. Culture media were stored at 4 °C and used for duration of 2 weeks. PBMCs were thawed carefully to avoid loss of cell viability and functionality. Samples were removed from liquid nitrogen and placed on ice, after which they were thawed in a 37 °C water bath. Once thawed, 500 μl warm complete RPMI medium supplemented with 10% FBS, 1% penicillin-streptomycin, and L-glutamine was added to cryovials dropwise. The cells were then transferred to 15 ml polystyrene tubes containing 10 ml warm medium and centrifuged for 10 min at $400 \times g$ to wash off the toxic DMSO. This wash step was repeated, and the cell pellet was then resuspended in medium to count as before. Once thawed, PBMCs were rested overnight to remove any apoptotic cells. After resting period, cells were washed to be used in culture. PBMCs were plated in 96 well plates using complete RPMI-1640 medium (10% FBS), and incubated at 37 °C in a humidified, 5% $CO_2$ atmosphere. Cells were treated with human T-cell activator CD3/CD28 magnetic beads (2 μl: 80,000 cells) for 12 days in presence of IL6 10 ng/ml, IL1β 10 ng/ml, TGFβ 10 ng/ml, and IL23 10 ng/ml. Culture medium was changed every other day for the whole duration. Compounds were added to the polarized cells on day 14 for 48 h. On the measurement day, CD3/CD28 beads were removed, and cells were treated with cell stimulation cocktail for 5 h at the end of 48 h incubation. Plates were centrifuged at $400 \times g$ for 10 min. Supernatant was collected for further testing and cells were resuspended in 1x PBS and washed twice at $400 \times g$ before staining for surface and intracellular markers. PBMCs cell suspension were stained cells with live/dead$_{455\,(UV)}$ viability dye in 1:1000 concentration in 1x PBS. Cells were incubated in dye solution for 30 min in 4 °C. Control wells were used where 50% of the cells in those wells were killed by heat (65 °C for 1 min) and then returned to their respective wells and stained similarly for 30 min. Cells were then centrifuged at $400 \times g$ for 10 min and the dye solution was discarded. Cells were washed twice with 1x PBS before surface staining. After the last wash, supernatant was discarded, and plates were pulse vortexed to completely dissociate the pellets. Anti-human anti-CD3$_{BV510}$, anti-CD4$_{Alexa Fluor 488}$ and anti-CD8$_{PE-Cy7}$ were added to each well at 1:1000 concentration in 1% BSA in PBS. Cells were incubated for 1 h at 4 °C. Cells were then centrifuged at $400 \times g$ and washed with 1x PBS twice. After the last wash, supernatant was discarded, and plates were pulse vortexed to dissociate the pellets. Cells were treated with FOXP3 fixation/permeabilization buffer (100 μl/well) for 30 min at 4 °C. Plates were centrifuged at $400 \times g$ and buffer was discarded followed by two washing steps with FOXP3 permeabilization buffer (200 μl) and centrifugation at $400 \times g$. Anti-human anti-IL17A$_{PE}$, anti-RORγt$_{BV650}$, anti-IL21$_{APC}$, anti-IL22$_{PerCP-eFluor 710}$ and anti-IFNγ$_{APC-eFluor 780}$ were added to each well used at 1:1000 concentration in permeabilization buffer. Cells were incubated for 1 h at 4 °C, centrifuged at $400 \times g$ and washed with 1x PBS twice. Cell quantification, viability and intracellular cytokines

expression were analyzed by flow cytometry using BD LSR-Fortessa flow cytometer.

## Human Th1 polarization

PBMCs were plated in 96 well plates using complete RPMI-1640 medium (10% FBS), and incubated at 37 °C in a humidified, 5% $CO_2$ atmosphere. Cells were treated with human T-cell activator CD3/CD28 magnetic beads (2 μl: 80,000 cells) for 3 days in presence of IL2 (5 ng/ml). Culture medium was changed every other day for the whole duration and compounds were added to each well from day zero for 3 days. On the measurement day, CD3/CD28 beads were removed, and cells were treated with cell stimulation cocktail for 5 h. Plates were centrifuged at $400 \times g$ for 10 min. Cells were resuspended in 1x PBS and washed twice at $400 \times g$ before staining for surface and intracellular markers. PBMCs cell suspensions were stained cells with live/dead$_{455\,(UV)}$ viability dye, anti-human anti-CD3$_{BV510}$, anti-CD4$_{Alexa Fluor 488}$, anti-CD8$_{PE-Cy7}$, anti-IL17A$_{PE}$ and anti-IFNγ$_{APC-eFluor 780}$. Cell quantification, viability and intracellular cytokines expression were analyzed by flow cytometry using BD LSR-Fortessa flow cytometer.

## Human Treg polarization

PBMCs were plated in 96 well plates using complete RPMI-1640 medium (10% FBS), and incubated at 37 °C in a humidified, 5% $CO_2$ atmosphere. Cells were treated with human T-cell activator CD3/CD28 magnetic beads (2 μl: 80,000 cells) for 6 days in presence of IL2 (5 ng/ml) and TGFβ (5 ng/ml). Culture medium was changed every other day for the whole duration and compounds were added to each well from day zero for 6 days. On the measurement day, CD3/CD28 beads were removed, and cells were treated with cell stimulation cocktail for 5 h. Plates were centrifuged at $400 \times g$ for 10 min. Cells were resuspended in 1x PBS and washed twice at $400 \times g$ before staining for surface and intracellular markers. PBMCs cell suspension were stained cells with live/dead$_{455\,(UV)}$ viability dye, anti-human antibodies for cell surface antigens namely anti-CD3$_{BV510}$, anti-CD4$_{Alexa Fluor 488}$, anti-CD8$_{PE-Cy7}$, anti-CD25$_{BV786}$ and CD127$_{PE-Cy5}$, anti-IL17A$_{PE}$, anti-IFNγ$_{APC-eFluor 780}$ and FOXP3$_{PerCP-Cy5.5}$ Cell quantification, viability and intracellular cytokines expression were analyzed by flow cytometry using BD LSR-Fortessa flow cytometer.

## RORγt TR-FRET binding assay

Five μl of compounds solutions in assay buffer were added to 15 μl of detection mix for 20 μl total assay volume (50 mM tris-HCl pH 7.0, 150 mM NaCl, 50 mM KCl, 5 mM MgCl2, 1 mM DTT, 0.1% BSA, 0.001% triton X); 1.5 nM GST-RORγt (LBD) (Creative BioMart, Shirley, NY, USA); 90 nM biotinylated RIP140 coactivator peptide (biotinyl-NH-Ahx-NSHQKVTLLQLLLGHKNEEN-CONH2) (CPC scientific inc., Sunnyvale, CA, USA); 50 nM SA-APC (PerkinElmer, Waltham, MA, USA); 1.5 nM Eu-anti GST IgG (PerkinElmer, Waltham, MA, USA); 1% DMSO). Compound dilutions (200x in pure DMSO) were prepared by making a 0.5 mM dilution from a 10 mM stock using 100% DMSO. Serial dilutions of the compounds were then prepared for 11 points beyond the 20 μM starting concentration in assay buffer. A 4x solutions containing europium-labeled antibody, and GST-RORγt (LBD) mixture were prepared, then 5 μl of each solution was added to each well on a 384-well plate, followed by the addition of 5 μl of each concentration of a compound previously diluted in assay buffer (described above). Assay controls (0% and 100% inhibition controls) were added into columns 1, 2, 23, and 24 of the 384 well assay plate. For 0% inhibition control, DMSO in assay buffer was used, while for 100% inhibition control, GSK2981278 (100 nM final concentration) was added. The plate was shaken for 1 min, centrifuged at 1000 rpm for 10 s, incubated at 4 °C overnight and followed by the addition of 5 μl of 4x biotinylated RIP140 peptide & streptavidin-APC, and read on a plate reader. For assay screening, Enspire plate reader was used. Settings were as follows: excitation: 320 nm, emission A: 615 nm; time delay: 220 μs; window: 600 μs, emission B: 699 nm (delay time 400 μs, window 800 μs); number of flashes: 100. RORγt TR-FRET assay is an end

point assay with a readout (emission ratio) of acceptor/donor multiplied by 10,000. The assay dose response testing is performed in duplicate points per concentration, with ten dilution concentrations per compound curve. The conversion of raw data to % activity is performed using assay controls, where 100% activity is represented by the average DMSO controls. Zero percent activity is the average of two wells of 100 nM, GSK2981278 compound controls. Inhibitory concentration curve fitting is performed using the sigmoidal dose-response (variable slope) equation as follows: $Y = 100/(1 + 10^{\wedge}((\log IC50-X) * hillslope))$ where $X$ is the logarithm of concentration and $Y$ is the normalized response. $Y$ begins at the bottom (0%) and goes to top (100%) with a sigmoid shape.

### Sensitized mouse model of skin transplantation

The animal protocol (MUHC-8128) was approved by the Animal Care Committee of the Research Institute of McGill University Health Centre. C57BL/6 mice were sensitized by IP injections of allogenic splenocytes ($1 \times 10^7$ on day 0, 7, and 14) derived from spleens of donor BALB/C mice[50]. This procedure was performed for 14 days following the initial sensitization step. BALB/C donor dorsal skin was transplanted to C57BL/6 recipient. The technique is described in detail by Cheng et al.[48]. BALB/C donor mice were anesthetized with isoflurane (induction vaporizer at 4%, maintenance at 1% through the mouse cone). Depth of anesthesia was monitored using toe pinch withdrawal reflex. The back of the donor mouse was shaved with electric razor and the skin was disinfected with 10% povidone iodine. Skin (from hip to neck) was harvested aseptically with blunt dissection at the level of the areolar connective tissue. Following skin harvest, animals were euthanized by cervical dislocation. Skin was separated from underlying panniculus carnosus layer, fat and connective tissues and cut out into squares (each square was used as a graft for an individual recipient mouse) of $12 \times 12$ mm size and stored on gauze with sterile PBS in petri dish. C57BL/6 mice were anesthetized with isoflurane (induction vaporizer at 4%, maintenance at 1% through the mouse cone). Depth of anesthesia was monitored using toe pinch withdrawal reflex. Buprenorphine (0.02 mg/kg) was administered for postoperative pain relief. The back of the donor mouse was shaved with electric razor and the skin was disinfected with 10% povidone iodine. Graft beds were created by cutting out $11 \times 11$ mm to $13 \times 13$ mm square of skin, while preserving the underlying layers intact (i.e., panniculus carnosus). Grafts from BALB/C donor were placed on the graft beds and were sutured avoiding folding at the edges and across the panniculus carnosus. Following transplantation mice were left to recover partially from anesthesia and surgery. Drugs were administered either by intraperitoneal injection or applied topically to skin grafts. Topical administration, 0.1% compound ointment (hydrous lanolin) or sham were applied to the skin graft and covered with hydrogel dressing and wrapped in an adhesive bandage with folded gauze. Transplanted mice were placed in a clean cage over a microwavable heating pad until fully recovered from anesthesia and for at least 1 h before returning to the housing facility. Bandages were changed every other day and grafts were observed and clinical scores were given to each. Mice which show signs of scabbing, contraction, or hardness in the first 72 h were considered a technical failure and were excluded from the study. The severity of rejection was measured using a scoring function from 0 to 5. Lesions were photographed using a digital camera and were evaluated by a blinded investigator[51]. Score 5 was considered the end point for each animal ($\geq$ 95% of the graft tissue becomes necrotic). At the end of rejection, animals were euthanized by cardiac puncture and blood was collected for further analysis. Skin grafts were harvested for histology. Spleens were collected for testing by flow cytometry. Animals were sacrificed on rejection day. ASEPESIS wound score for mice was used to evaluate surgical wound and rejection was scored according to graft area percent necrosis where complete rejection was >95% necrosis[27,51].

### Mouse T-cell to Th17 polarization

Splenocytes were isolated from mouse spleens using red blood cell lysis buffer and cell strainer[52]. Splenocytes were frozen in a medium containing 10% DMSO and 90% fetal bovine serum (FBS) and were placed inside a freezing container at −80 °C overnight to allow gradual and even cooling. The following day, samples were moved to a liquid nitrogen tank for long-term storage. Samples were removed from liquid nitrogen and placed on ice, after which they were thawed in a 37 °C water bath. Once thawed, 500 µl warm complete RPMI medium supplemented with 10% FBS, 1% penicillin-streptomycin, and L-glutamine was added to each cryovial dropwise. The cells were then transferred to 15 ml polystyrene tubes containing 10 ml warm medium and centrifuged for 10 min at $400 \times g$ to wash off the toxic DMSO. This wash step was repeated, and the cell pellet was then resuspended in medium to count cells. T cells were isolated from −80 frozen mouse splenocytes using T cell isolation cocktail and magnetic streptavidin beads[52]. Following the isolation, cells were suspended in PBS and counted. Mouse T cells were plated in 96 well plates using complete RPMI-1640 medium (10% FBS), and incubated at 37 °C in a humidified, 5% CO2 atmosphere. Cells were treated with mouse T-cell activator CD3/CD28 magnetic beads (2 µl: 80,000 cells) for 4 days in presence of IL6 (20 ng/ml), IL1β (10 ng/ml), TGFβ (2 ng/ml), and IL23 (10 ng/ml). Compounds were added to the polarized cells on day zero for the whole duration of the polarization. On the measurement day, CD3/CD28 beads were removed, and cells were treated with cell stimulation cocktail for 5 h. Plates were centrifuged at $400 \times g$ for 10 min. T cells were stained cells with live/dead$_{455 (UV)}$ viability dye, anti-mouse anti-CD4$_{Alexa Fluor 488}$, anti-CD8$_{PE-Cy7}$, anti-IL17A$_{PE}$, RORγt$_{BV650}$, anti-IL21$_{APC}$, anti-IL22$_{PerCP-eFluor 710}$ and anti-IFNγ$_{APC-eFluor 780}$. Cell quantification, viability and intracellular cytokines expression were analyzed by flow cytometry using BD LSR-Fortessa flow cytometer.

### Mouse splenocytes immunostaining

B cells: live/dead$_{eFluor 780}$, anti-CD3$_{BV510}$, anti-CD19$_{PE-Cy7}$, anti-CD138$_{PerCP-Cy5.5}$, anti-IgG3$_{Alexa Fluor 594}$, anti-IgM$_{Alexa Fluor 350}$; ILC3 cells: live/dead$_{eFluor 506}$, anti-CD4$_{Alexa Fluor 488}$, anti-CD8$_{PE-Cy7}$, anti-CD45$_{APC-Cy7}$, anti-CD117$_{APC}$, anti-CD127$_{PerCP-eFluor 710}$, anti-Lin$_{eFluor 450}$; T cells: live/dead$_{455 (UV)}$, anti-CD4$_{Alexa Fluor 488}$, anti-CD8$_{PE-Cy7}$, anti-IL17A$_{PE}$, anti-RORγt$_{BV650}$, anti-IL21$_{APC}$, anti-IL22$_{PerCP-eFluor 710}$, anti-IFNγ$_{APC-eFluor 780}$; Macrophages: live/dead$_{455 (UV)}$, anti-CD169$_{Alexa Fluor 488}$, anti-IL17A$_{PE}$, anti-RORγt$_{BV650}$, anti-IL21$_{APC}$, anti-IL22$_{PerCP-eFluor 710}$, anti-IFNγ$_{APC-eFluor 780}$; Neutrophils: live/dead$_{455 (UV)}$, anti-Ly6G$_{FITC}$, anti-IL17A$_{PE}$, anti-RORγt$_{BV650}$, anti-IL21$_{APC}$, anti-IL22$_{PerCP-eFluor 710}$, anti-IFNγ$_{APC-eFluor 780}$.

### Statistics and reproducibility

Statistical analysis was performed using GraphPad Prism (version 10.1.2) (GraphPad Software, SanDiego, CA, USA) with the suitable statistical test as indicated in each analysis. $p$ Values were classified as not significant (ns; $p > 0.05$), significant ($*0.01 < p < 0.05$), very significant ($**0.001 < p < 0.01$) or extremely significant ($***p < 0.001$ and $****p < 0.0001$). Data are presented as mean ± standard deviation (SD). Scoring of skin grafts rejection or histology were done independently by the study authors. Independent evaluations of the grafts were done by veterinarians and veterinary technicians within their settings and events of graft rejections were reported to authors. Blinding was carried out to all authors and support scientists who participated in the study. Additional experiments were carried out to confirm results as necessary.

### Reporting summary

Further information on research design is available in the Nature Portfolio Reporting Summary linked to this article.

### Data availability

All data supporting the findings of this study are available within the paper and its Supplementary Information. The supplier catalog numbers, and other relevant details of the antibodies, kits and other materials used in the study can be found in Supplementary Tables 1–8. Source data for the graphs are provided in Supplementary Data 1–5.

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

## Acknowledgements

The authors thank Dr. Jean Claude Bertrand, Drug Discovery Platform, Research Institute of McGill University Health Centre, Montréal, Québec, Canada for his unwavering support and guidance throughout the research process. The authors also thank Dr. Jean-Jacques Lebrun and Dr. Jonathan Cools-Lartigue, Research Institute of McGill university Health Centre, Montréal, Québec, Canada for their invaluable feedback on the study results. The authors are grateful to the team of the Immunophenotyping Platform, Research Institute of McGill University Health Centre, Montréal, Québec, Canada. The authors thank the team of Animal Resources Division, Research Institute of McGill University Health Centre, Montréal, Québec, Canada. Finally, the authors thank Fazila Chouiali and Mohamed Djallali, Histopathology Platform, Research Institute of McGill University Health Centre, Montréal, Québec, Canada.

## Author contributions

A.F. wrote the manuscript, designed, and conducted all the experiment and data analysis. M.M. helped with rejection scoring and daily dosing. A.K. helped with mouse surgeries. S.N. helped with flow cytometry experiments. S.P. and J.T. helped with the study design, manuscript writing and mouse surgeries.

## Competing interests

The authors declare no competing interests.
