## [Peer Review File · Communications Biology]

Reviewers' comments:

Reviewer #1 (Remarks to the Author):

In organ transplantation, pre-existing antibodies against given leukocyte and HLA antigens are used as a predictive factor of chronic graft rejection. The patients bearing those antibodies at high levels are defined "sensitized". In this paper, Fouda et al. initially showed a link between the sensitized state and the increase of Th17 cells in human blood samples. They additionally showed that the Th17 polarization in sensitized patients is repressed by a novel ROR γ t inverse agonist (TF-S14) which was recently generated by the authors themselves (Fouda et al., 2023 J Med Chem, PMID 37172324). The authors then tested the efficacy of the compound in a mouse model where the mice were sensitized with alloantigens prior to transplantation with skin allograft. The paper provides an important insight to the clinical application of ROR γ t-targeting compounds, but some improvements are required to enforce their conclusion and make the presentation more reader-friendly.

Specific comments:

- 1) Before jumping to the allosensitization and transplantation model (Figure 3 and later), the authors need to clearly show the effect of TF-S14 on the helper T cell polarization in vitro using mouse T cells.
- 2) The allosensitization mouse model would be unfamiliar to the most of the readers. Enough background knowledge on this model should be provided in the text, especially the detail of underlying immune reaction against the graft. In case of normal skin allograft transplantation model (without sensitization treatment), allo-reactive CD8 T cells are primarily associated with the acute rejection. Are CD8 T cells not important in the sensitization model? Can you exclude CD8 T cells from the possible targets of the compound?
- 3) Representative FACS profiles for figures 1 and 2 should be presented in the supplemental materials, so the readers can judge the biological validity of the data.
- 4) Figures 4D and E (immunohistochemistry) should be presented with lower magnification. Additionally, numerical analysis of positively stained cells should be done on these data.

Reviewer #2 (Remarks to the Author):

The authors established a sensitized murine skin allograft model, and treated with a novel ROR γ t inverse agonist TF-S14 followed by detections of IL-17-producing cells, B cell differentiation and antibody class switching, et al. The results showed that TF-S14 reduced IL-17A production in Th17 and leukocytes, and blocked the differentiation of B cells, and finally prolonged the survival of a total mismatch skin grafts. This study is well-designed, data are well-organized, and conclusions are sound. The authors present a novel ROR γ t inverse agonist TF-S14, which offers a therapeutic intervention through a novel mechanism to treat rejection in highly sensitized patients.

Minor comments:

1. In Fig.3G, "TF-S14, 100 mg/kg" lost
2. For the Fig. 3E-G, the medium survival time (MST) of skin grafts in each group should presented in Result section.
3. In Fig. 5, the legend for 5C lost.

Reviewer #1 (Remarks to the Author):

In organ transplantation, pre-existing antibodies against given leukocyte and HLA antigens are used as a predictive factor of chronic graft rejection. The patients bearing those antibodies at high levels are defined “sensitized”. In this paper, Fouda et al. initially showed a link between the sensitized state and the increase of Th17 cells in human blood samples. They additionally showed that the Th17 polarization in sensitized patients is repressed by a novel ROR γ t inverse agonist (TF-S14) which was recently generated by the authors themselves (Fouda et al., 2023 J Med Chem, PMID 37172324). The authors then tested the efficacy of the compound in a mouse model where the mice were sensitized with alloantigens prior to transplantation with skin allograft. The paper provides an important insight to the clinical application of ROR γ t-targeting compounds, but some improvements are required to enforce their conclusion and make the presentation more reader-friendly.

Specific comments:

1) Before jumping to the allosensitization and transplantation model (Figure 3 and later), the authors need to clearly show the effect of TF-S14 on the helper T cell polarization in vitro using mouse T cells.

Answer: Thank you for the suggestion. I added four subfigures under Figure 3 as part of validating the animal model and at the same time validate the effect of TF-S14 on T cell polarization.

2) The allosensitization mouse model would be unfamiliar to the most of the readers. Enough background knowledge on this model should be provided in the text, especially the detail of underlying immune reaction against the graft. In case of normal skin allograft transplantation model (without sensitization treatment), allo-reactive CD8 T cells are primarily associated with the acute rejection. Are CD8 T cells not important in the sensitization model? Can you exclude CD8 T cells from the possible targets of the compound?

Answer: Thank you for the suggestion. We benefited from the reviewer’s advice to spread the message to a wider audience and added three sentences in the introduction section to describe the models in a concise language yet accessible to most readers. In addition, we discussed some examples of the animal models under the discussion section as needed. With regards to CD8, we have some data, however we don’t have a conclusion and we will need to do another animal study that is suited for CD8 and acute cellular rejection. The model we are presenting is rather a hyperacute and it known as a “white graft” which onset is 24 hours as the antibodies against graft is already “performed”. While acute rejection is cellular and requires two weeks to reach 95-100% necrosis.

3) Representative FACS profiles for figures 1 and 2 should be presented in the supplemental materials, so the readers can judge the biological validity of the data.

Answer: We attached the required supplementary figures in a separate word file with FACS plots for Figure 1 and 2. In addition, we added two more supplementary figures, one for the polarization experiments and another one for Immunohistochemistry.

4) Figures 4D and E (immunohistochemistry) should be presented with lower magnification. Additionally, numerical analysis of positively stained cells should be done on these data.

Answer: We performed the numerical analysis based on one color staining and moved the dual staining figure to supplementary section. Additionally, we added another figure in supplementary section to emphasize on the cooperation between IL17A+ cells and neutrophils that takes place inside the graft and leads to rejection.

Reviewer #2 (Remarks to the Author):

The authors established a sensitized murine skin allograft model, and treated with a novel ROR γ t inverse agonist TF-S14 followed by detections of IL-17-producing cells, B cell differentiation and antibody class switching, et al. The results showed that TF-S14 reduced IL-17A production in Th17 and leukocytes, and blocked the differentiation of B cells, and finally prolonged the survival of a total mismatch skin grafts.

This study is well-designed, data are well-organized, and conclusions are sounded. The authors present a novel ROR γ t inverse agonist TF-S14, which offers a therapeutic intervention through a novel mechanism to treat rejection in highly sensitized patients.

Minor comments:

1. In Fig.3G, "TF-S14, 100 mg/kg" lost

Answer. Thank you we corrected the figure as instructed.

2. For Fig. 3E-G, the median survival time (MST) of skin grafts in each group should be presented in Results section.

Answer. Thank you for your comment we have added the MST for each of the mice groups in the results section.

3. In Fig. 5, the legend for 5C lost.

Answer. Thank you for highlighting this. We corrected the letter numbering.

REVIEWERS' COMMENTS:

Reviewer #1 (Remarks to the Author):

Authors properly reponded to the comments from the reviewers.

Reviewer #2 (Remarks to the Author):

All my concerns have been well issued in the Manuscript-R1. Thanks.